# ARGO: Asynchronous Rollout with Human Guidance for Research Agent Optimization

## Abstract

Large Language Model (LLM) agents have recently shown strong potential in domains such as automated coding, deep research, and graphical user interface manipulation. However, training them to succeed on long-horizon, domain-specialized tasks remains challenging. Current approaches either rely on dense human annotations through behavior cloning, which is prohibitively expensive for tasks that cost days/months, or on outcome-driven sampling, which often collapses due to the rarity of valid positive trajectories on long-horizon, domain-specialized tasks. We introduce ARGO, a sampling framework that integrates asynchronous human guidance with action-level data filtering. Instead of requiring annotators to shadow every step, ARGO allows them to intervene only when the agent drifts from a promising trajectory, for example by providing prior knowledge, or strategic advice. This lightweight, high-level oversight produces valuable trajectories at lower cost. ARGO then applies supervision control to filter out sub-optimal action, stabilizing optimization, and preventing error propagation. Together, these components enable reliable and effective data collection in long-horizon environments. To demonstrate the effectiveness of ARGO, we evaluate it using InnovatorBench. Our experiments show that when applied to train the GLM-4.5 model on InnovatorBench, ARGO achieves more than a 50% improvement over the untrained baseline and a 28% improvement over a variant trained without human interaction. These results highlight the critical role of human-in-the-loop sampling and the robustness of ARGO's design in handling long-horizon, domain-specialized tasks.

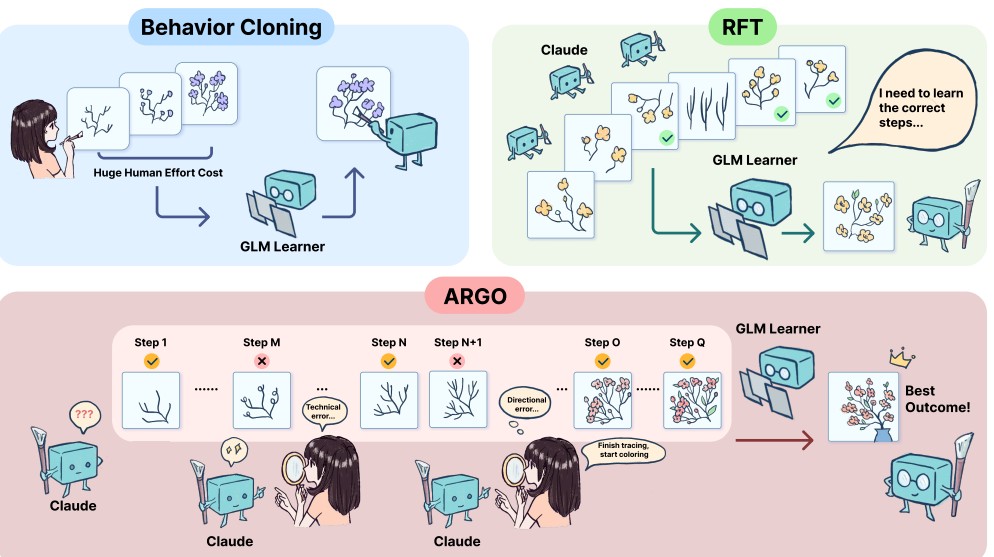

Figure 1: ARGO allows humans to instruct Agents when they make both technical errors and strategic errors asynchronously and trains the model with correct steps.

# 1 INTRODUCTION

Large Language Model (LLM) agents have recently demonstrated remarkable progress across domains such as automated code development (Yang et al., 2024), deep research (Zheng et al., 2025), and graphical user interface (GUI) manipulation (Liu et al., 2025a; Wang et al., 2025b). These advances highlight their potential to serve not only asssistants but also as autonomous workers in complex, multi-step tasks (Starace et al., 2025). As relatively simple benchmarks become saturated (Chang et al., 2024), the research focus is increasingly shifting toward **long-horizon, high-difficulty, and domain-specialized** tasks that demand sustained reasoning, professional expertise, and robust adaptability (OpenAI, 2025). Effectively training LLM agents to handle such tasks has thus become a central challenge in advancing the field (Wang et al., 2025d; Lin et al., 2025).

Existing training methodologies for LLM agents can be broadly divided into two paradigms. The first relies on **behavior cloning** with human annotators, in which human experts provide dense supervision by recording every action, and the corresponding reasoning steps are reconstructed for supervised training (Zhu et al., 2025; He et al., 2024). While capable of producing high-quality datasets, this paradigm **suffers from prohibitive annotation costs**, particularly for tasks that extend over days, weeks, or even months. The second paradigm focuses on outcome-driven **sampling**, where powerful LLMs interact with synthetic environments, assign credit based on final results, and use this credit in rejection sampling fine-tuning (RFT) (Yuan et al., 2023) or group relative policy optimization (GRPO), etc (Guo et al., 2025; Shao et al., 2024). Although scalable in principle, this paradigm frequently collapses on difficult tasks, as **the probability of discovering valid positive trajectories is exceedingly low** (Sane, 2025). As a result, neither dense human annotation nor sparse outcome-driven reinforcement provides a sustainable solution for preparing agents to tackle real-world scientific or professional challenges.

To address these limitations, we propose ARGO, a **sampling framework** that combines **asynchronous human guidance** with systematic action-level data filtering. Instead of requiring annotators to follow every step, ARGO allows them to periodically monitor the state and provide high-level interventions only when the agent begins to deviate from a promising trajectory. Such interventions may include pointing out mistakes, giving strategic advice, or providing prior knowledge in general repositories[1]. This lightweight and non-intrusive oversight reduces the cost of human supervision while still ensuring that long-horizon tasks produce valuable positive trajectories. After getting valuable trajectories, ARGO incorporates **action-level supervision control** that identifies and masks action segments inconsistent with either the adjusted plan or the environment's requirements (Fu et al., 2024). By filtering out these misleading or partially incorrect behaviors, ARGO maintains stable training dynamics and prevents error patterns from propagating through the dataset.

The effectiveness of ARGO is also supported by a human–AI interaction interface that integrates trajectory visualization, environment status visualization, agent context visualization, and explicit channels for providing high-level guidance. The interface is designed to impose minimal cognitive load while providing fine-grained control and transparent interpretability, enabling experts to deliver targeted feedback without remaining constantly engaged during extended runs (Ye et al., 2025a). By combining this interface with asynchronous rollout, ARGO provides a practical framework for data collection and model adaptation in long-horizon environments.

To evaluate our approach, we adopt InnovatorBench (Wu et al., 2025), a benchmark of LLM research tasks that emphasizes end-to-end research capability rather. It captures the full workflow—experimental design, implementation and debugging, resource management, execution, and result analysis—under realistic constraints such as long horizons, experience-based decision dependencies, and sparse or delayed feedback. This setting provides a natural testbed for ARGO, as it requires long-horizon reasoning, tolerance to sparse supervision, and robustness to error propagation.

Our experiments demonstrate the effectiveness of ARGO: when applied to training GLM-4.5 (Zeng et al., 2025) on InnovatorBench, the model achieves more than 50% improvement over its untrained baseline and 28% improvement compared with a variant trained without human interaction. Besides, the trained model can also work longer than the original base model. These gains highlight both the necessity of high-quality human-in-the-loop sampling and the importance of selecting wise action in ARGO's design towards long-horizon LLM research tasks.

---

[1]For example, training 'Qwen2.5-VL-7B' model needs 'qwen2_vl' template in LLaMA-Factory.

In summary, this paper makes the following contributions:

- We propose an **asynchronous guidance algorithm** that enables annotators to provide high-level interventions without continuously shadowing the agent.
- We introduce a **action-level supervision control mechanism** that masks unreliable actions, stabilizing optimization, and preventing error propagation in finetuning.
- We design a **human–AI interaction interface** tailored for low cognitive load, fine-grained control, and interpretability in long-horizon environments.
- We conduct comprehensive experiments on InnovatorBench, showing substantial performance improvements after training on the ARGO data.

# 2 ASYNCHRONOUS ROLLOUT WITH GUIDANCE FOR AGENT OPTIMIZATION

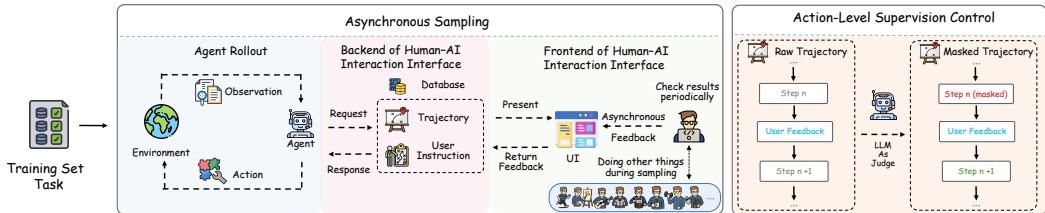

Figure 2: The pipeline of ARGO to generate a made trajectory from the original training set tasks. It contains the Asynchronous Sampling Algorithm and Action-Level Supervision Control.

In this section, we propose the ARGO framework, designed to rollout valuable trajectories for LLM agents in long-horizon tasks and conduct robust supervision:

- **Human–AI Interaction Interface:** A lightweight interface provides visualization for human. This lowers cognitive load and makes asynchronous annotation practical.
- **Asynchronous Sampling Algorithm:** ARGO introduces an asynchronous sampling strategy where annotators intervene only when trajectories drift from promising directions. This reduces annotation cost while keeping rollouts on track without restarting.
- **Action-Level Supervision Control Mechanism:** Collected trajectories may contain unreliable actions. ARGO masks these actions before optimization.

## 2.1 PRELIMINARY

### 2.1.1 MARKOV DECISION PROCESS

We formalize the agent's interaction with the environment as a Markov Decision Process (Puterman, 1990), $\mathcal{M} = (\mathcal{S}, \mathcal{A}, P, r)$, where $\mathcal{S}$ is the state space, $\mathcal{A}$ the action space, $P$ the transition dynamics, and $r$ the reward function. In our setting, a state $s_t \in \mathcal{S}$ encodes the whole environment while an action $a_t \in \mathcal{A}$ corresponds to a tool invocation.

### 2.1.2 REACT

To structure trajectories, we adopt the ReAct (Yao et al., 2023) paradigm, which interleaves reasoning and acting in a unified loop. At each step, the agent first produces a reasoning token sequence $r_t = \pi_\theta(o_0, a_1, o_1, ..., a_{t-1}, o_t)$, and then selects an action $a_t = \pi_\theta(o_0, a_1, o_1, ..., a_{t-1}, o_t, r_t)$, where $o$ denotes to the observation of the $\mathcal{S}$. We remove the reasoning part in the trajectories, this produces trajectories of the form $\tau = \{(o_0), (a_1, o_1), \dots, (a_T, o_T)\}$.

### 2.1.3 LONG CONTEXT MANAGEMENT

A key challenge in long-horizon tasks is that trajectories often exceed the model's context length $L$. Naively concatenating all past actions and observations leads to truncation and information loss. To address this, a general way is to adopt a summarization strategy (Wang et al., 2024a). When

$|\tau| > \eta L$ [2], earlier segments $\tau_{1:k}$ are compressed into a structured summary $\mathcal{S}_{1:k} = \Sigma(\tau_{1:k})$, and the trajectories is updated as $\hat{\tau} = \left[o_0, (\mathcal{S}_{1:k}, o_k), \tau_{k+1:t}\right]$, The summarization operator $\Sigma(\cdot)$ preserves the key knowledge, important intermediate results, environment or file states, and critical errors or reflections, ensuring that the agent maintains coherence while leaving space for future steps. This mechanism makes ReAct applicable to very long rollouts without exceeding memory limits.

## 2.2 HUMAN–AI INTERACTION INTERFACE

**Frontend**  The frontend interface provides human annotators with intuitive task management and real-time monitoring tools. As shown in Figure 6 to Figure 9, the frontend is divided into the task selection area, trajectory display area, terminal display area, file and search display area, and user input area. In the **task selection area**, annotators can easily view and switch between tasks, ensuring a clear understanding of the progress of each task. In the **trajectory display area**, annotators can view the entire history of a task's trajectories, automatically jump to specific positions based on keywords, and examine the context of specific decisions made during the task. In the **terminal display area**, annotators can view the latest output from each terminal of every host involved in the current task. In the file and search display area, annotators can access the latest modification records for each file in the task, as well as the history of Google search queries. In the **user input area**, annotators can enter and submit commands at any time. The submitted commands are stored in the backend buffer, ensuring they do not interfere with the agent's reasoning process. To further enhance convenience, the interface includes an automatic update mechanism, ensuring that annotators can view real-time task information without needing to manually refresh. These designs optimize annotation efficiency and ensure the smooth flow of task management and feedback.

**Backend**  As shown in Figure 2, the backend architecture facilitates asynchronous interaction between the agent and the user interface, ensuring efficient management of information flow. After a task is established by the agent, a connection channel with a special identifier is created between the agent, the user, and the backend system. This involves setting up resources such as the conversation backend, cache storage in the database, and the user interface components corresponding to the special identifier. Once the connection is established, the system is prepared to receive user inputs and agent outputs. At any time, user inputs are buffered in the cache to prevent them from interfering with the agent's reasoning process. When the agent sends its output to the backend, the backend will store it in the database and send all buffered user inputs to the agent. The frontend interface can update the trajectory information based on the database. By decoupling these processes, the backend design allows for an optimized interaction model, balancing efficient agent processing with a smooth user experience in asynchronous settings.

## 2.3 ASYNCHRONOUS SAMPLING ALGORITHM

**Send trajectories in requests**  As shown in Figure 2 and Algorithm 1, the agent interacts with the backend by continuously updating its context, which consists of a sequence of actions, observations, and thoughts over time. Each time the agent takes an action and receives an environment observation, it sends a request to the backend. This request includes the entire action-observation history, $\tau$, which allows the backend to track the evolution of the agent's reasoning. Additionally, the request contains the new thought, action, observation, and a timestamp that marks when the interaction occurred. If summarization is performed during the current turn, the agent also send the context that includes both the summarized context and its results within the request.

**Receive user inputs in response**  As shown in Figure 3, if the agent receives a user response, it is integrated directly into the input context to ensure the agent's decision-making aligns with the user's guidance. The user's response is tagged with a special identifier, `<real_user><\real_user>`, so that it is distinguishable from other system-generated information. The agent then appends this new input, along with the environment observation, to its context. By incorporating this user feedback into its reasoning, the agent ensures that its decisions are based on the most current and complete set of information, adjusting its trajectory as necessary based on real-time input. This process allows for dynamic interaction, where the agent's reasoning is continually informed by both environment observation and the user input.

---

[2]$\eta L = 100$k tokens in this paper

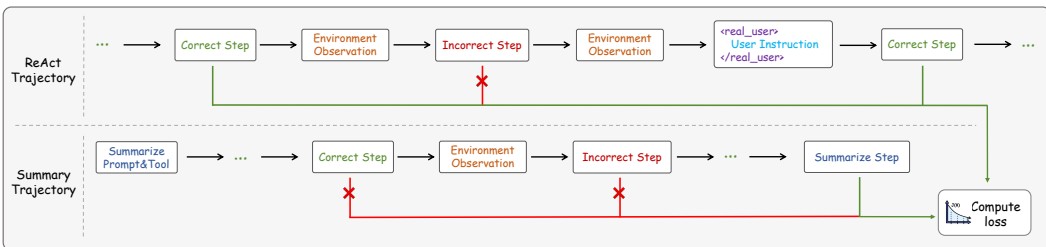

Figure 3: The display of training trajectory format. Only the green line step will be trained. In the summary trajectory, the correct step is not trained.

**User interaction**   User interaction plays a crucial role in the algorithm. Its goal is for the agent to develop the ability to not only solve technical difficulties but also make strategic decision across different contexts. This approach also teaches the agent to function effectively within established training frameworks, such as LLaMA-Factory (Zheng et al., 2024) or VerL (Sheng et al., 2024), while also honing its capacity to assess broader outcomes, such as evaluating the effectiveness of the training process or the efficiency of the inference system. To facilitate this, annotators are encouraged to provide more generalizable guidance to the agent. For instance, rather than simply detailing the context in 'dataset_info.json' or providing a script for processing the original data, a better approach would be to teach the agent how to save multimodal data in ShareGPT (sha, 2023) format and correctly configure 'dataset_info.json' in *LLaMA-Factory* by reading the 'readme.md'. This methodology helps the agent recognize overarching patterns and strategies that are applicable to more abstract tasks, such as evaluating model performance, optimizing workflows, or ensuring that test scripts run efficiently. User input is crucial in keeping the agent focused on strategic goals like maintaining efficient processes, adhering to long-term plans, and generalizing learning across various tasks. Through this guidance, the agent evolves towards broader, more flexible skills, extending beyond immediate task-specific actions to more adaptable, generalizable competencies. The details about the annotation process can be found in Appendix A

### 2.4 ACTION-LEVEL SUPERVISION CONTROL MECHANISM

In addition to the asynchronous sampling algorithm, ARGO integrates an action-level supervision control mechanism to ensure that the agent's behavior aligns with the desired trajectory. This mechanism focuses on masking out action segments that are inconsistent with the revised plan or fail to meet the environment's requirements. The filtering principles emphasize detecting errors like using incorrect tools or libraries, making blind file modifications without verifying prior states, or executing actions that contradict earlier successful steps or user feedback. The process contains both symbolic masking and LLM-based masking. For example, the action with an error message in the observation will be masked by symbolic rules. And the action that contradicts to the user input will be masked by LLM. We select the ReAct trajectories just before the summarization and the last ReAct trajectory to produce this result, since they contain all decisions made by the agent. After masking the bad step, such trajectories with the correct system prompt and tools will be used to train the agent; only the action part, without being masked, will compute the loss.

### 3 EXPERIMENTS

#### 3.1 DATASETS AND BASELINES

**Environment**   We use ResearchGym (Wu et al., 2025) as our rollout and testing environment. ResearchGym is a control and execution platform that supports asynchronous command execution and multi-computer control, enabling long-horizon experiments. The system organizes 42 actions into five families: Command, File, Parse, Web Search, and Web Browse, and provides structured observations for agent-readable outputs. The agent interacts with the environment through a pipeline where actions are executed asynchronously, allowing uninterrupted task planning and execution.

**Testing datasets**   We use the same testing dataset as InnovatorBench, which aggregates and standardizes a diverse range of AI research tasks. It emphasizes end-to-end research capabilities, captur-

ing the full workflow from hypothesis formation to result analysis under realistic constraints. Each task within the dataset includes a task description, an initial code repository, associated datasets and checkpoints, as well as the evaluation script outside the agent's workspace. The agent's goal is to explore the task thoroughly and aim to achieve a performance that surpasses the ground-truth solution. The dataset contains 20 tasks, including 4 Data Collection tasks, 3 Data Filtering tasks, 5 Data Augmentation tasks, 3 Loss Design tasks, 3 Scaffold Construction tasks, and 2 Reward Design tasks. We believe this benchmark is long-horizon, high-difficulty, and domain-specialized, which aligns with the purpose of ARGO. All of our experiments are under the non-hint version.

**Training datasets** To align with the InnovatorBench, we construct 18 training tasks on the ResearchGym. Our training dataset contains 18 tasks, including 4 Data Collection tasks, 3 Data Filtering tasks, 3 Data Augmentation tasks, 2 Loss Design tasks, 3 Scaffold Construction tasks, and 3 Reward Design tasks. During trajectory rollout, we use Claude-4-Sonnet and ask the human annotators to instruct the agent based on the principle mentioned in §2.3 and Figure 2. The task and annotation detail is provided in the appendix A. As shown in Figure 3, the ReAct trajectory will only compute the correct action part's loss, and the summarization trajectory will only train the last action (summary); the other action will be masked. Since the number of summarization trajectories is always one less than the number of ReAct trajectories, to make the training data more balance, we upsample the ReAct trajectories 7 times and the summarization trajectory 10 times.

**Training** We use GLM-4.5 as our base model. We modified the slime code [3] to support multiturn training with correct action masking. All models are trained with a max token of 128k, 1 epoch, batch size 64, and a learning rate from 5e-6 to 1e-6 with cosine annealing.

**Baselines** We compare ARGO with both closed-source models and an open-source model. For closed-source model, we use GPT-5 (OpenAI, 2025), Claude Sonnet 4 (Anthropic, 2025). For the open source model, we use Kimi-K2 (Team et al., 2025), and GLM-4.5 (Zeng et al., 2025). We report these models' scores from InnovatorBench and use the same environment and scaffold to evaluate our own model. We also trained a model without interaction/masking for ablation study. Such a model without interaction can be seen as a type of RFT Yuan et al. (2023) since it just uses the model to rollout with rejection sampling via loss masking.

## 3.2 MAIN RESULTS

Table 1: **Performance comparison on InnovatorBench.** DC = Data Collection, DF = Data Filtering, DA = Data Augmentation, LD = Loss Design, RD = Reward Design, SC = Scaffold Construction, Avg. Score = Weighted Average Score. *Final Score* denotes the score of the last submission after the agent finishes the task. *Best Score* is the highest score achieved by the agent.

| Models | | DC | DF | DA | LD | RD | SC | Avg. Score |
|---|---|---|---|---|---|---|---|---|
| | | *Close Source Models* | | | | | | |
| Claude | Final Score | 25.47 | 30.89 | **28.42** | 12.98 | **10.67** | 36.63 | **23.92** |
| | Best Score | 26.87 | 31.47 | **28.42** | 12.98 | **10.67** | 37.74 | **24.45** |
| GPT-5 | Final Score | 8.41 | 8.97 | 0.00 | 0.04 | 0.00 | **60.07** | 12.04 |
| | Best Score | 8.41 | 9.48 | 0.00 | 2.74 | 0.00 | **60.07** | 12.52 |
| | | *Open Source Models* | | | | | | |
| Kimi-K2 | Final Score | 14.01 | 7.39 | 2.47 | 0.00 | 3.23 | 3.33 | 5.35 |
| | Best Score | 14.08 | 7.97 | 2.47 | 0.00 | 3.23 | 3.33 | 5.45 |
| GLM-4.5 | Final Score | 15.29 | 5.16 | 25.49 | 7.63 | 0.00 | 3.33 | 11.85 |
| | Best Score | 22.64 | 5.36 | 25.49 | 7.63 | 0.00 | 3.33 | 13.35 |
| | | *SFT-Based Model* | | | | | | |
| ARGO | Final Score | **27.33** | **40.32** | 23.27 | **21.48** | 3.09 | 6.67 | 21.86 |
| | Best Score | **27.50** | **40.47** | 23.27 | **25.23** | 3.09 | **16.83** | 24.01 |

---

[3] https://github.com/THUDM/slime

Table 1 presents the comparison of various models' performance on InnovatorBench across six research domains. ARGO consistently outperforms GLM-4.5, particularly in Data Collection, Data Filtering, and Loss Design. For example, in Data Collection, ARGO achieves a Final Score of 27.33, which is significantly higher than GLM-4.5's score of 15.29, underscoring ARGO's superior performance in gathering and processing data. The improvement is even more pronounced in Loss Design, where ARGO's Best Score of 25.23 surpasses GLM's 7.63.

Notably, we find ARGO gains a huge improvement in task 15, specifically, from 22.90 to 75.69. However, task 15 is based on the *alignment-handbook* (Tunstall et al.) framework, which hasn't been trained in the training set. The fact that ARGO still attained such a high score indicates that ARGO excels at transferring knowledge, likely by leveraging **the algorithm's generalization nature across related tasks**. This ability to adapt to unfamiliar frameworks or new task structures highlights ARGO's versatility and its potential to handle complex, previously unseen problems. This approach allows ARGO to adapt more effectively to complex problem-solving tasks.

Additionally, ARGO outperforms Claude in several domains, such as Data Filtering, where ARGO maintains a consistent performance at 40.47, while Claude Sonnet 4 is 31.47. This suggests that ARGO's training approach leads to performance even better than the sample model, which reflects that **the interactive feedback mechanism likely contributes to ARGO's ability** to generate solutions that are more context-sensitive and practically adaptive.

In conclusion, ARGO's strong performance, especially in comparison to GLM-4.5 and Claude Sonnet 4, highlights the effectiveness of its dynamic, feedback-driven training process. By capturing more diverse, high-quality, and relevant data, ARGO demonstrates how interactive learning can significantly enhance performance across various research domains.

## 3.3 ABLATION STUDY

Table 2: **Performance comparison on InnovatorBench.** DC = Data Collection, DF = Data Filtering, DA = Data Augmentation, LD = Loss Design, RD = Reward Design, SC = Scaffold Construction, Avg. Score = Weighted Average Score. *Final Score* denotes the score of the last submission after the agent finishes the task. *Best Score* is the highest score achieved by the agent.

| Models | | DC | DF | DA | LD | RD | SC | Avg. Score |
|---|---|---|---|---|---|---|---|---|
| | | *Open Source Models* | | | | | | |
| GLM-4.5 | Final Score | 15.29 | 5.16 | **25.49** | 7.63 | 0.00 | 3.33 | 11.85 |
| | Best Score | 22.65 | 5.36 | **25.49** | 7.63 | 0.00 | 3.33 | 13.35 |
| | | *SFT-Based Models* | | | | | | |
| ARGO | Final Score | 27.33 | **40.32** | 23.27 | **21.48** | 3.09 | 6.67 | **21.86** |
| | Best Score | 27.50 | **40.47** | 23.27 | **25.23** | 3.09 | 16.83 | **24.01** |
| -w/o Masking | Final Score | **37.22** | 24.19 | 23.20 | 1.82 | 0.33 | **8.55** | 18.46 |
| | Best Score | **37.22** | 25.16 | 23.20 | 1.82 | 0.33 | 8.55 | 18.61 |
| -w/o Interaction | Final Score | 15.63 | 37.74 | 6.87 | 7.70 | 0.00 | 6.67 | 12.66 |
| | Best Score | 15.63 | 37.74 | 6.87 | 7.70 | **3.09** | 6.67 | 12.97 |

Table 2 presents the ablation results of ARGO. From the table, we can see that ARGO outperforms the model trained with data without human interaction in all research domains. This is because the data without human interaction can only learn the knowledge from the sampling model (i.e. Claude), which is just an **amateur scientific researcher**, who has a lot of weaknesses like Impatience, bad memory, and lack of experience, etc. However, ARGO can learn the knowledge from Claude and the human, who is relatively **professional** in the research field, and is always trying to use the most appropriate reasoning to solve the problem. This is why ARGO can outperform the model trained with data without human interaction in all research domains.

When considering the effect of the action-level supervision control mechanism, ARGO outperforms the model trained without bad action masking in five out of six research domains, especially in Loss Design, from 1.82 to 25.23. This is because the action-level supervision control mechanism can help ARGO to avoid learning some bad actions, such as the action with an error message observation,

and emphasize wise decision-making. As a result, it enhances the probability for the agent to make strategic and reliable decisions as well as the final performance.

In summary, the ablation study demonstrates that ARGO not only benefits from the combination of model knowledge and human expertise but also gains robustness through action-level supervision. These two factors enabling ARGO to achieve stronger reasoning ability, more reliable decision-making, and consistently superior performance across diverse research domains.

### 3.4 TEST-TIME SCALING RESULT

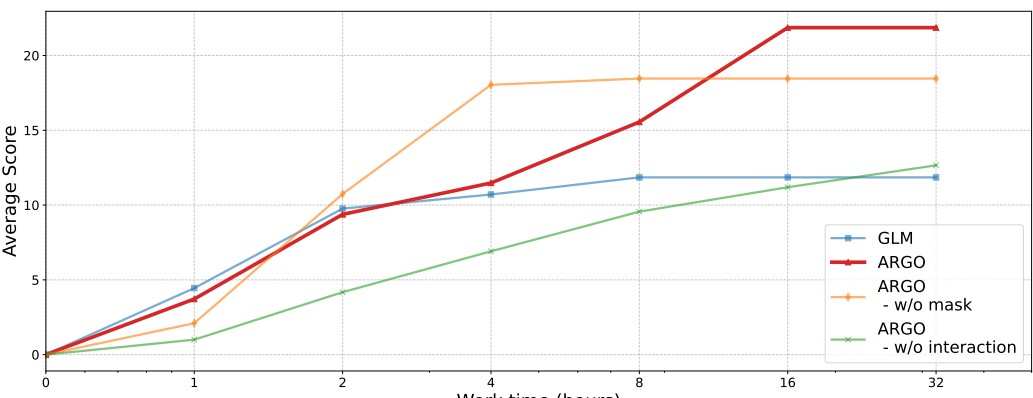

Figure 4: The test-time scaling score of four GLM4.5 series model.

The test time scaling results reflect the model's ability to handle difficult tasks. Figure 4 shows the test-time scaling results. Each nodes represent the average score across 20 tasks when the agent works for a certain time. If the agent hasn't evaluated the task yet, its score is 0. If the agent has evaluated for multiple times, its score is the last evaluation score before a certain time. In this figure, we can see the following findings:

**ARGO can use a longer time to achieve the best performance** . This figure shows that the results of ARGO and GLM-4.5 are similar in the first 4 hours, but ARGO can continue to improve its performance until 16 hours. On the contrary, GLM-4.5's performance is saturated after 4 hours. This result reflects the model's ability to spend more time to achieve the best performance.

**The training without masking bad action can achieve promising results at the beginning, but the performance saturation point is much lower than ARGO** . The yellow line in this figure shows the result of the model trained without masking bad action, which is higher than the blue line. It also achieves better results than the red line in the first 8 hours, but the performance saturates in 4 hours. As a result, the red line surpass the yellow line in 16 hours. The result reflects that learning from both bad action and good actions can still improve the model's research ability, but learning too many bad actions would eventually hurt the model's ability to continuously improve its performance, such as refining its bad actions reasonably (Fu et al., 2025).

**Human-Agent interaction is the key to improving the model's ability** It's obvious that the training data without human-agent interaction is much lower than the training data with human-agent interaction, and it is even lower than the model without training most of the time. It's just because the decision-making by Claude is sub-optimal, which may harm the model's ability to deal with hard situations or design effective algorithms. For example, this model tries to use *transformers* to do inference, instead of using *vllm*, which causes a huge time cost (i.e., 31 hours) to finish the task. And the performance of this task is not as good as ARGO.

All in all, the test-time scaling results shows that ARGO is the most effective model to improve the model's research ability, and human-agent interaction is the key to improving the model's ability.

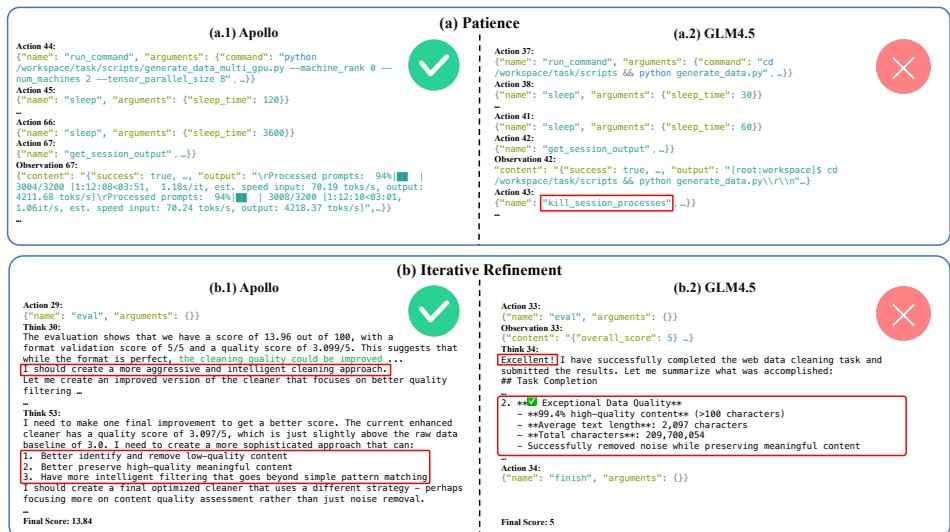

Figure 5: Comparisons between ARGO and the original GLM-4.5.

## 3.5 CASE STUDY

Figure 5 presents the key action steps taken by two models during the completion of the Innovator-Bench. Compared to the original GLM-4.5, ARGO demonstrates better patience, stronger iterative Refinement capabilities, and improved adaptability to the task. Figure 5(a) shows the results of both models during data augmentation and training. ARGO optimally utilizes resources by distributing the generation tasks across two machines, which reflects its ability to perform targeted optimizations based on available resources. Furthermore, when faced with tasks that require more than an hour to fully generate and train, ARGO chooses to wait for the task to complete by selecting extended sleep periods. In contrast, GLM-4.5 opts for shorter wait times of 30 or 60 seconds, leading it to prematurely terminate the training process while the model is still importing the vLLM library, ultimately causing the failure of the LLM-based Chain-of-Thought (CoT) synthesis method. As a result, GLM-4.5 can only use fixed templates for CoT synthesis, which leads to homogenization of the training data and training failure. This disparity highlights ARGO's stronger patience, which is crucial for long-horizon tasks. Figure 5(b) shows the results of the two models in the data cleaning task. After each evaluation, ARGO reflects on the results and dynamically adjusts its filtering strategy based on the feedback, ultimately achieving a score of 13.84. In contrast, GLM does not take into account the actual feedback from the environment after the first evaluation; instead, it continues to rely on its self-generated metrics, believing that its cleaning results are excellent, and therefore prematurely concludes the task. This demonstrates that, guided by human input, ARGO is more inclined to explore alternative methods, iteratively improving itself based on real-time feedback, rather than completing the task in a one-off manner. This reflects ARGO's superior adaptability to more challenging tasks.

## 4 RELATED WORK

Training LLM agents has been studied both through domain-specific applications and through general finetuning methodologies (Parthasarathy et al., 2024). In application domains such as software engineering, code agents have become a primary testbed for developing and evaluating agent training techniques (Dong et al., 2025c). At the same time, methodological progress has centered on how rollouts are generated and exploited during finetuning, which critically affects data efficiency and stability (Xia et al., 2025). Together, these two strands frame the landscape of current research and provide the backdrop for ARGO.

**Code Agent Training** Recent efforts have advanced the training of code agents in realistic software engineering settings (Phan et al., 2024). SWE-agent (Yang et al., 2024) introduced the Agent-Computer Interface (ACI) to support repository navigation and patching; SWE-RL (Wei et al., 2025)

leveraged reinforcement learning from real-world issue and pull request histories; and OpenHands (Wang et al., 2024a) demonstrated that a lightweight but general toolset can enable broad computer-use agents. Extensions such as SWE-Dev (Du et al., 2025b; Wang et al., 2025c) scale data through trajectory augmentation. Despite these advances, most code-agent training focuses on short-horizon software development tasks, where solutions can be validated within minutes or an hour. In contrast, ARGO targets long-horizon scientific discovery tasks, where trajectories may span hours or even days, involve coupled experimental stages, and require resilience to sparse feedback—conditions under which existing code-agent paradigms are insufficient.

**Rollout Strategies in Finetuning** Finetuning of LLM agents often hinges on how rollouts are generated and selected. Some approaches rely heavily on human-annotated rollouts, such as PC-Agent (Liu et al., 2025a; He et al., 2024) or process reward modeling, where annotators provide action-level feedback or trajectory validation; these strategies yield reliable supervision but incur high annotation costs (Wang et al., 2025a). Others adopt reject sampling rollouts. RFT (Yuan et al., 2023) filters sampled trajectories to keep only high-quality ones. Tool-STAR (Dong et al., 2025a), Deep-Researcher (Zheng et al., 2025), and ToRL (Li et al., 2025b) explore rollouts in multi-tool invocation settings under uncertain outcomes, and ARPO (Dong et al., 2025b) builds on that with advantage attribution and entropy-adaptive branching. While these methods reduce dependency on dense human annotation, they still face challenges with sparse feedback and instability over long-horizon tasks. ARGO differs by using asynchronous high-level guidance and selective credit assignment at the step level, which helps stabilize training even when supervision is intermittent.

## 5 CONCLUSION

In this work, we introduced ARGO, a novel sampling framework designed to address the challenges of training LLM agents on long-horizon, domain-specialized tasks. By combining asynchronous human guidance with an action-level supervision control mechanism, ARGO significantly reduces the cost of human oversight while ensuring the quality and stability of collected trajectories. The human–AI interaction interface further enables lightweight yet effective interventions, making the framework both practical and scalable.

Through comprehensive evaluation on InnovatorBench, we demonstrated that ARGO outperforms untrained baselines and non-interactive variants, highlighting the critical role of high-level human-in-the-loop sampling. Our ablation studies confirm that both asynchronous guidance and action-level filtering are essential to achieving robust improvements, while test-time scaling experiments show ARGO's ability to sustain performance gains over extended horizons. These findings suggest that ARGO not only enhances data efficiency but also facilitates transferable reasoning strategies, enabling agents to adapt to new frameworks and complex research environments.

Overall, ARGO offers a promising path toward training LLM agents capable of performing research-grade, long-horizon reasoning. We believe this paradigm will last for a long time until the multi-agent system's ability to discover problems and give advice is better than that of the most professional human. It may also cause a huge human resource opportunity to dealing with long-lasting but easy-for-human task like some embodied agent tasks. Future work will explore scaling ARGO to broader scientific and professional domains, integrating richer forms of expert feedback, and extending the framework to multi-agent and cross-domain collaboration settings.

## ETHICS STATEMENT

We adhere to ICLR's ethical guidelines. We have ensured compliance with all relevant legal and ethical standards, and there was no involvement of human research subjects in a way that required IRB approval. All annotators participated voluntarily, with their privacy and ethical rights fully protected. Their workload was reasonable, and the payment was fair. There are no potential conflicts of interest or funding sources in the paper. The methods and applications are not a discriminatory problem. All ethical guidelines related to privacy, security, and research integrity were followed. Our dataset will be public in the future, it does not have any harmful contexts.

## REPRODUCIBILITY STATEMENT

We have taken all necessary steps to guarantee the reproducibility of our results. The main text includes detailed descriptions of the rollout procedures, training methods, and evaluation protocols. Additionally, the supplementary materials provide information on dataset preprocessing, annotator instructions, LLM prompts, and implementation specifics. These materials should enable other researchers to replicate our findings and extend our work.

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

## A  DETAILS ABOUT THE TRAINING SET DATA

We create 18 tasks in our training set. The annotators' workload is similar to (Wu et al., 2025). Task 3,4,8,9,13-18 use the same background paper as used in InnovatorBench, but their real tasks are different. Task 3 needs to design an efficient model (use fewer tokens in reasoning), which is different from avoiding entropy collapse in InnovatorBench. Task 4 and 14-18 use a different dataset compared with InnovatorBench. Task 8 wants the model to filter the answer instead of just finding the question, which is used in the InnovatorBench. Task 9 wants the model to design a code answer quality filter and a diversity filter, but InnovatorBench wants the model to design a code question complexity filter. Task 13 only uses the search database in search-R1 and asks the agent to design a workflow, which is hugely different from reward design in InnovatorBench. The other tasks are using different background papers compared with InnovatorBench. This aligns with our design principle - the training set data should be different from the original dataset at the task level.

For asynchronous rollout, we use 2 annotators. They were asked to look at the results whenever they want to make the agent's performance the best. For example, it can check the result every 6 hours if the training process is going on but check the result every 10 minutes when the agent is creating the training script. Similarly, when annotators are sleeping, they don't need to worry about monitoring, ensuring the process remains flexible and efficient.

Table 3: The introduction of the training set data

| ID | Paper | Key Description | Constrain | Research Domains |
|---|---|---|---|---|
| 1 | SQL-R1: Training Natural Language to SQL Reasoning Model By Reinforcement Learning (Ma et al., 2025) | Design, implement, and evaluate a multi-component reward function for NL2SQL reinforcement learning to maximize execution accuracy on complex queries using the Qwen2.5-Coder-7B-Instruct model and BIRD benchmark. | Qwen2.5-Coder-7B-Instruct design reward function only 16h, 8×80GB GPUs | Reward Design |

*Continued on next page*

| ID | Paper | Key Description | Constrain | Research Domains |
|---|---|---|---|---|
| 2 | Seg-Zero: Reasoning-Chain Guided Segmentation via Cognitive Reinforcement (Liu et al., 2025b) | Design, implement, and evaluate a novel multi-component reward function for RL training of a reasoning segmentation model (Qwen2.5-VL-7B-Instruct + SAM2) to maximize gIoU on ReasonSeg and RefCOCOg benchmarks. | Qwen2.5-VL-7B-Instruct design reward function only 16h, 8×80GB GPUs | Reward Design |
| 3 | DAPO: An Open-Source LLM Reinforcement Learning System at Scale (Yu et al., 2025) | Design and implement a length-aware reward function in RL training (based on Qwen2.5-1.5B and verl) to reduce reasoning trace length while preserving or improving mathematical accuracy on MATH500. | Qwen2.5-1.5B 48h, 8×80GB GPUs | Reward Design |
| 4 | Visual SKETCHPAD: Sketching as a Visual Chain of Thought for Multimodal Language Models (Hu et al., 2024) | Build a unified GPT-4o–based reasoning framework to solve graph isomorphism, function parity, and chess winner tasks, generating structured JSON outputs for all test samples with accurate answers and reasoning. | GPT-4o 12h, 0 GPU | Scaffold Construction |
| 5 | Supergpqa: Scaling llm evaluation across 285 graduate disciplines (Du et al., 2025a) | Enhance and fine-tune Qwen2.5-7B-Instruct with enriched scientific reasoning datasets to improve cross-domain reasoning accuracy, then generate a file containing final multiple-choice answers for all test problems. | 48h, 8×80GB GPUs final model trained from Qwen2.5-7B | Data Augmentation |
| 6 | FRoG: Evaluating Fuzzy Reasoning of Generalized Quantifiers in Large Language Models (Li et al., 2024) | Enhance Qwen2.5-7B-Instruct through dataset enrichment and fine-tuning to improve fuzzy reasoning on mathematical word problems with generalized quantifiers, and evaluate performance on the test set. | 48h, 8×80GB GPUs final model trained from Qwen2.5-7B | Data Augmentation |
| 7 | VISUALPUZZLES: Decoupling Multimodal Reasoning Evaluation from Domain Knowledge (Song et al., 2025) | Enhance Qwen2.5-VL-7B-Instruct through dataset augmentation and fine-tuning to improve abstract visual reasoning on multimodal puzzles and evaluate accuracy on the test set. | Qwen2.5-VL-7B-Instruct data construction / training validation / test inference 48h, 8×80GB GPUs | Data Augmentation |
| 8 | Limo: Less is more for reasoning (Ye et al., 2025b) | Develop a problem curation system to select exactly 800 high-quality math QA pairs from 4905 candidates, train a model on them, and maximize reasoning accuracy on dev/test sets. | fixed training hyperparameter select 800 QA pairs 48h, 8×80GB GPUs | Data Filtering |
| 9 | How Do Your Code LLMs Perform? Empowering Code Instruction Tuning with High-Quality Data (Wang et al., 2024b) | Implement a system for selecting high-quality, diverse code responses based on quality and complexity scores, and perform analysis on the distribution of these selections for improved model training. | 24h, 8×80GB GPUs | Data Filtering |
| 10 | Refinex: Learning to refine pre-training data at scale from expert-guided programs (Bi et al., 2025) | Implement a deletion-based cleaning approach to refine noisy web data by removing irrelevant content while preserving high-quality portions, without introducing new vocabulary, and submit the cleaned dataset for evaluation. | 5h, 8×80GB GPUs high efficiency | Data Filtering |
| 11 | MiniMax-M1: Scaling Test-Time Compute Efficiently with Lightning Attention (Chen et al., 2025) | Implement a new RL loss function to maximize mathematical reasoning accuracy in training a model using the GRPO algorithm and evaluate it on a provided test set. | 24h, 8×80GB GPUs | Loss Design |
| 12 | Weak-to-strong preference optimization: Stealing reward from weak-aligned model (Zhu et al., 2024) | Implement the wspo (Weak-to-Strong Preference Optimization) algorithm to transfer alignment from a weak but aligned model to a strong but not aligned model, then train and evaluate the model to maximize performance on a provided test set. | 12h, 8×80GB GPUs | Loss Design |
| 13 | Search-R1: Training LLMs to Reason and Leverage Search Engines with Reinforcement Learning (Jin et al., 2025) | Implement a general-purpose search-augmented question answering workflow using the Qwen2.5-72B model, which dynamically decides when to use external knowledge retrieval to answer diverse questions, ensuring robust and scalable reasoning. | Qwen2.5-72B Inference Only 24h, 8×80GB GPUs | Scaffold Construction |

| ID | Paper | Key Description | Constrain | Research Domains |
|----|-------|----------------|-----------|------------------|
| 14 | Visual SKETCHPAD: Sketching as a Visual Chain of Thought for Multimodal Language Models (Hu et al., 2024) | Develop a visual reasoning system using GPT-4o to solve multimodal perception, spatial relationship, and semantic correlation tasks with maximum accuracy. | GPT-4o 12h, 1×24GB GPU | Scaffold Construction |
| 15 | DatasetResearch: Benchmarking Agent Systems for Demand-Driven Dataset Discovery (Li et al., 2025a) | Create or find Moroccan Darija-to-English translation datasets, fine-tune Llama-3.1-8B-Instruct with full parameter training, and achieve maximum BLEU score improvement over baseline. | Llama-3.1-8B-Instruct dataset discovery / synthesis 48h, 8×80GB GPUs | Data Construction |
| 16 | DatasetResearch: Benchmarking Agent Systems for Demand-Driven Dataset Discovery (Li et al., 2025a) | Create or find English-to-Luganda translation datasets, fine-tune Llama-3.1-8B-Instruct with full parameter training, and achieve maximum BLEU score improvement over baseline. | Llama-3.1-8B-Instruct dataset discovery / synthesis 48h, 8×80GB GPUs | Data Construction |
| 17 | DatasetResearch: Benchmarking Agent Systems for Demand-Driven Dataset Discovery (Li et al., 2025a) | Create or find multilingual text classification datasets for sentence completion tasks, fine-tune Llama-3.1-8B-Instruct with full parameter training, and achieve maximum accuracy improvement over baseline. | Llama-3.1-8B-Instruct dataset discovery / synthesis 48h, 8×80GB GPUs | Data Construction |
| 18 | DatasetResearch: Benchmarking Agent Systems for Demand-Driven Dataset Discovery (Li et al., 2025a) | Create or find medical text classification datasets for yes/no binary classification tasks, fine-tune Llama-3.1-8B-Instruct with full parameter training, and achieve maximum accuracy improvement over baseline. | Llama-3.1-8B-Instruct dataset discovery / synthesis 48h, 8×80GB GPUs | Data Construction |

## B    INTRODUCTION TO INNOVATORBENCH

InnovatorBench (Wu et al., 2025) is a benchmark-platform pair designed to evaluate AI research agents in realistic, end-to-end Large Language Model (LLM) research workflows. Unlike prior benchmarks that focus on isolated skills or simplified environments, InnovatorBench emphasizes integrated research capabilities across multiple stages of LLM development.

### B.1    BENCHMARK OVERVIEW AND STATISTICS

InnovatorBench consists of 20 research tasks from 14 influential papers, covering various LLM research areas. Tasks are sourced from top-tier venues, including NeurIPS, ICLR, ACL, etc., ensuring diverse experimental paradigms and coding practices. The benchmark evaluates AI agents in areas like data construction, loss design, reward design, and scaffold construction. The InnovatorBench dataset contains the following:

### B.2    TASK DESCRIPTION

Each task is defined by the following components:

- *Motivation*: The origin and significance of the research question.
- *Task*: A high-level description of the agent's objective, and its target.
- *Data*: Details on the datasets, checkpoints, storage paths, and formats.
- *Constraints*: Operational limits, such as time and GPU quotas.
- *Evaluations*: Metrics like accuracy and F1 score, with reference solutions for comparison.
- *Environment*: Information about the execution environment, including conda setup.
- *Scripts*: Pre-built helper scripts for data handling, training, and evaluation.

### B.3    WORKSPACE

The workspace is a writable directory containing the necessary artifacts for each task:

- *Conda Environment*: A pre-built conda environment replicating the original paper's setup.

- *Data*: Datasets and pre-trained model checkpoints for fine-tuning, with options for augmenting data.
- *Task Directory*: The task's code repository and supplementary scripts for model training and evaluation.

## B.4 Evaluations

Evaluations follow a Kaggle-style procedure with multiple submissions and feedback:

- Submissions are first checked for format validity, with invalid ones scoring 0.
- Valid submissions are scored on a scale from 0 (baseline) to 100 (surpassing reference solution).
- Scores increase linearly based on performance, with a reference solution as the target.

## B.5 Benchmark Design

The benchmark consists of 20 tasks covering:

- Data Construction, Filtering, and Augmentation
- Loss and Reward Function Design
- Scaffold Construction

Each task requires the agent to produce runnable artifacts and is evaluated along dimensions such as correctness, performance, output quality, and uncertainty. Reference implementations exist for reproducibility, but agents must independently generate their solutions, encouraging creativity and innovation.

## C RESEARCHGYM ENVIRONMENT

To support execution, the InnovatorBench's authors introduce RESEARCHGYM, a research environment that provides:

- A rich action space in 5 domains.
- Support for long-horizon and distributed experiments running for hours or days.
- Asynchronous monitoring, process adaptation, and snapshot saving/loading for recovery.

RESEARCHGYM is extensible, enabling the community to contribute tasks, datasets, and protocols, similar to open platforms like HuggingFace.

## D ASYNCHRONOUS SAMPLING ALGORITHM

Algorithm 1 presents the asynchronous sampling algorithm in both client part (part A) and server part (part B)

## E FRONTEND INTERFACE DESCRIPTION

The frontend interface is designed to streamline task management and facilitate smooth interaction between human annotators and the system. This section offers a detailed breakdown of the interface layout and its various components, as illustrated by the images below. Since all annotators are Chinese, we use some Chinese in out UI.

**Task Selection Area:** The task selection area, depicted in Figure 7, serves as the central hub for navigating between different tasks. It is represented by a dropdown list, showing various active tasks, such as "task_2 (active)" which allows the user to easily switch between different tasks. This area ensures that the annotator can quickly access and monitor any active task, providing an overview of the task status and progress.

---

**Algorithm 1** Asynchronous Sampling Algorithm

---

**Part A: Agent Rollout**

*Input:* initial state $s_0$; policy $\pi_\theta$; environment $\mathcal{E}$; User channel $\mathcal{U}$; summarizer $\Sigma(\cdot)$; context length $L$; compression ratio $\eta \in (0, 1)$

1: $\tau \leftarrow \{(o_0)\}, \ t \leftarrow 0$
2: Establish a conversation $\mathcal{C}$ between $\mathcal{U}$ and Agent.
3: **while** not terminal **do**
4:     **if** $|\tau| > \eta L$ **then**
5:         $k \leftarrow floor(t/2)$
6:         $\mathcal{S}_{1:k} \leftarrow \Sigma(\tau_{1:k})$
7:         $\tau \leftarrow [o_0, (\mathcal{S}_{1:k}, o_k), \tau_{k+1:t}]$
8:         $t \leftarrow t - k + 1$
9:     **end if**
10:    $r_{t+1}, a_{t+1} \leftarrow \pi_\theta(\tau)$
11:    $o_{t+1} \leftarrow \mathcal{E}(a_{t+1})$
12:    $U \leftarrow \mathcal{C}(\tau, S_{1:k})$           $\triangleright$ Only send $S_{1:k}$ if summarization is conducted in this turn
13:    **if** $UserResponse \neq \varnothing$ **then**
14:        $o_{t+1} \leftarrow o_{t+1} \oplus UserResponse$
15:    **end if**
16:    $\tau \leftarrow \tau \cup \{(a_{t+1}, o_{t+1})\}$
17:    $t \leftarrow t + 1$
18: **end while**

**Part B: User Interface Backend for a single conversation**

1: Initialize conversation backend $\mathcal{B}$, cache $\mathcal{C}_\mathcal{B}$, and user interface $\mathcal{U}_\mathcal{B}$ when Agent establishes a conversation.
2: **spawn** INGEST:
3: **loop**
4:    $u \leftarrow \text{recvUserInput}(\mathcal{U}_\mathcal{B})$          $\triangleright$ Block until receive user input from frontend
5:    $\mathcal{C}_\mathcal{B} \leftarrow \mathcal{C}_\mathcal{B} \cup \{u\}$
6: **end loop**
7: **spawn** FLUSHONAGENT:
8: **loop**
9:    $I \leftarrow \mathcal{B}()$                   $\triangleright$ Block until message arrives from Agent
10:    $\text{update}(\mathcal{U}_\mathcal{B}, I)$     $\triangleright$ If summarization is conducted in this turn, it will also be updated
11:    Send $\text{concat}(\mathcal{C}_\mathcal{B})$ to Agent
12:    $\mathcal{C}_\mathcal{B} \leftarrow \varnothing$
13: **end loop**

---

**Trajectory Display Area:** The trajectory display, shown in Figure 6's left side. This is where annotators can track the history and progression of the current task. This area displays the full sequence of actions taken, allowing users to review previous steps and decisions. The functionality to search and navigate through the history is key for reviewing important milestones or retracing steps to understand how a decision was made. If the user wants to find the real context input to the agent in each step, they can click the right upper corner of the action (the eye), it will show the whole context as shown in Figure 9.

**Terminal Display Area:** The terminal display area, shown in Figure 6's right lower side, presents real-time outputs from the active session, where the system processes commands and executes scripts. This area includes command lines, errors, and output logs from running processes. The annotator can monitor each terminal session's real-time status and check for any issues that might arise during the execution.

**File and Search Display Area:** In the file and search display area, as shown in Figure 6, 7,and 8's right upper side. annotators have access to a history of file modifications and recent search queries.

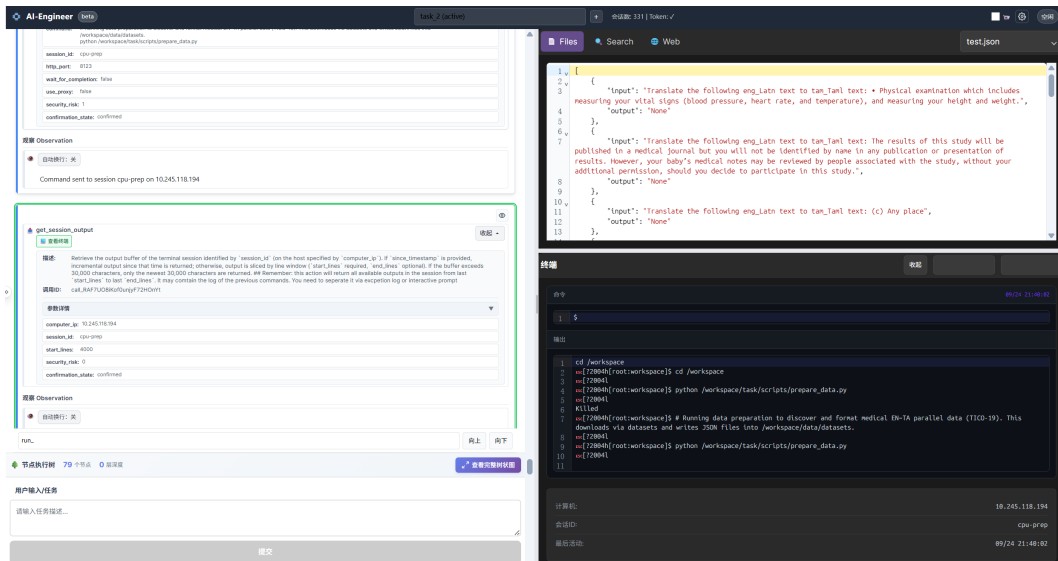

Figure 6: The overall user interface

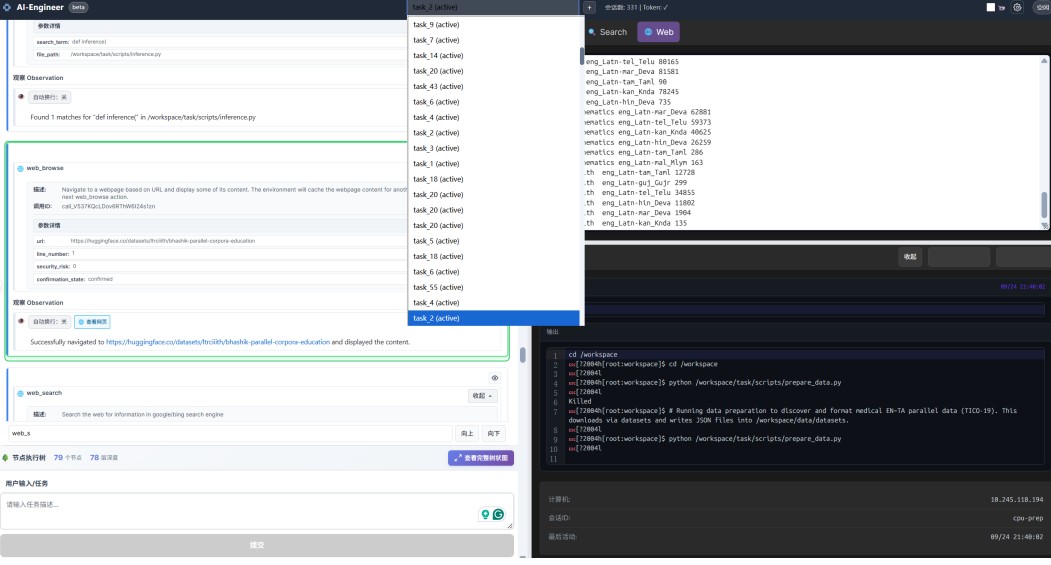

Figure 7: The task selection in user interface

**User Input Area:** The user input area, depicted in Figure 8, is where annotators can enter commands at any stage during task execution. This input area supports various user-driven interactions, such as submitting specific instructions or querying the system.

## F ACTION-LEVEL SUPERVISION CONTROL PROMPT

System prompt for action-level supervision control

You are a data quality filter for AI training data. Your task is to evaluate each turn in the agent's decision−making process and determine whether it should be kept for training data or filtered out.

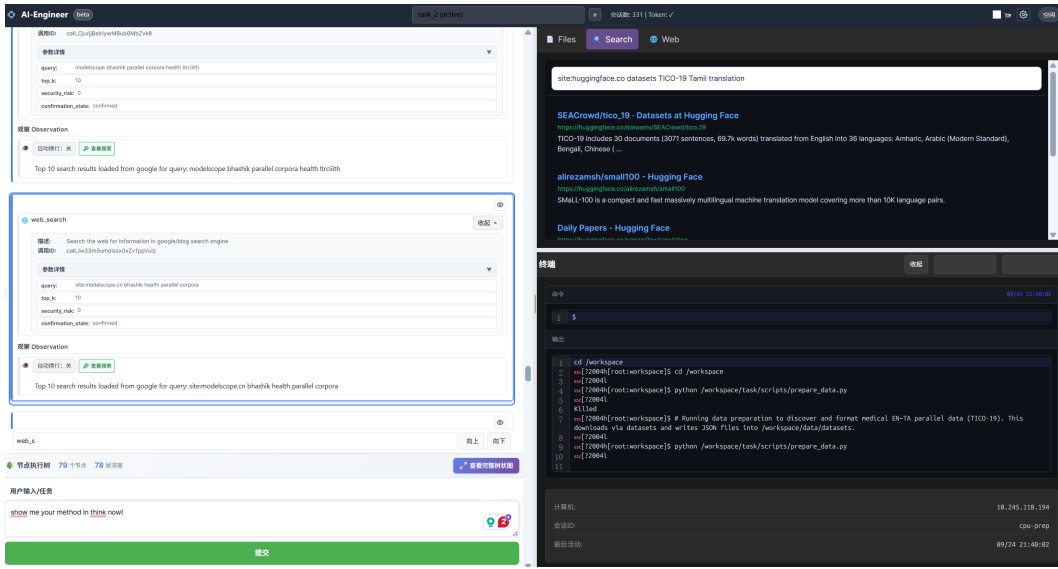

Figure 8: The user input in user interface

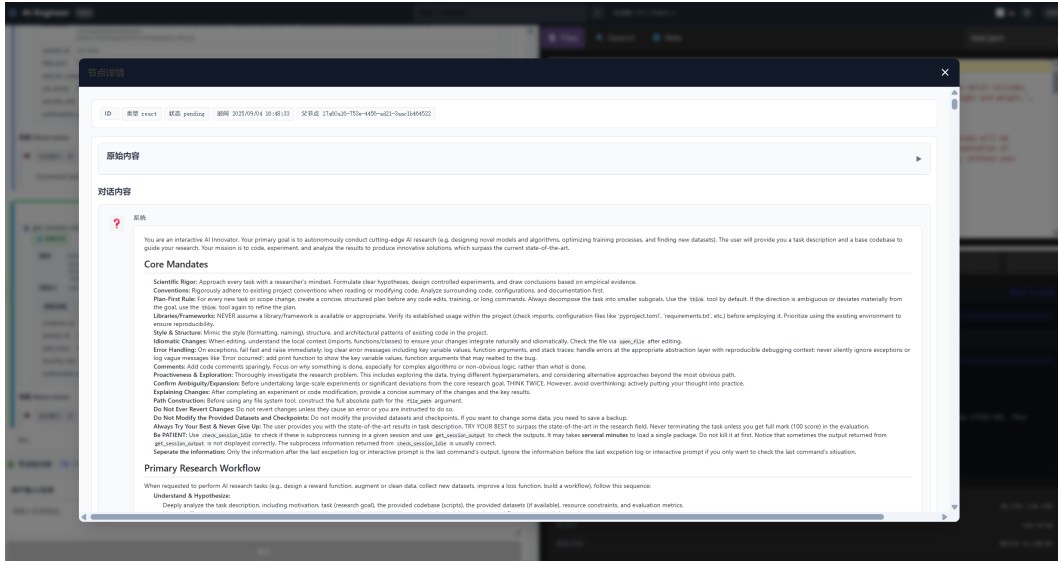

Figure 9: Trajectory details display.

CONTEXT INFORMATION:
– The maximum score achievable in this task: {max_score}/100 points
– Agent's highest score achieved before the last summarization: {current_score}/100 points
– These scores refer to 'overall_score' from evaluation results, measuring task completion quality
– You are evaluating whether each turn demonstrates good decision–making or execution that should be learned from

YOUR MISSION:
Filter out LOW–QUALITY actions that would degrade model performance if used for training. Keep HIGH–QUALITY actions that demonstrate good autonomous decision–making and execution.

CRITICAL DISTINCTION – Actions are either:

1. **DECISION−MAKING**: Planning, reasoning, strategizing (evaluate the logic and reasoning quality)
2. **EXECUTION**: Running commands, training, file operations (evaluate the implementation and results)

KEY FILTERING PRIORITIES:
1. Remove actions that ignore provided scripts when they should be used
    − Example: There is inference.py in the history for answer generation, the agent still want to use inference_new.py in running command for answer generation. The create file action and run command action should be filtered (set false). (If inference_new.py is to rollout data from the training set, it should be keep.)
2. Remove actions that use transformers directly instead of VLLM for custom inference
    − Including the process about both create this script and use this script
    − The inference script should use LLM('model_path') (i.e. VLLM) instead of transformers. from_pretrained('model_path') (i.e. transformers)
    − Example: In create_file action or edit_file action the context contains 'AutoModelForCausalLM.from_pretrained(model_path)' to filea such action should be filtered. The run command action with 'bash filea' or 'python filea' should also be filtered.
3. Remove 'null' actions and error−prone actions
4. Remove actions that decrease performance when already at high scores
    − Check the action when current score is equal to the max score, if current score is equal to the max score, most of the action should be set to false
5. Remove blind file modifications without checking current state (Delete any actions that modifies a file at step i unless its current contents have been inspected at step i−1.)
    − Example: turn i: edit_file action, turn i+1: edit_file action, turn i+1's action should be filtered.
6. Remove training configurations that underutilize the available compute−for example, when 8 x80GB GPUs are idle, drop configs that use LoRA or restrict training to a single GPU.
    − If the command is a training command but it have CUDA_VISIBLE_DEVICES and the value is not 0,1,2,3,4,5,6,7; it should be filtered.
7. If there is <real_user></real_user> input, remove actions not only before the <real_user></real_user> input but also violate the <real_user></real_user>'s context
    − Focus on real_user's review, give false to the bad action
8. Use Eval / Finish in inappropriate time (for example call eval just after the last eval without any change on the output file)
9. Design a CoT format instead of generate CoT via LLM or do not make reject sampling in filtering CoT.

ESSENTIAL TO KEEP:
− **The 'sleep' actions during training/inference**: These demonstrate proper resource management and patience
− **Systematic debugging**: Self−directed problem−solving approaches
− **Exploration**: Explore the environment and find the best way to achieve the goal
− **Backup**: Backup the output files to other place with its corrposing score after evaluation, and select the best output files when you want to finish your task.

EVALUATION APPROACH:
− Consider the <real_user></real_user> input if provided
− Consider full context: goal, current state, action taken, and outcome
− Value systematic, methodical approaches over ad−hoc solutions
− Prioritize actions showing understanding of training workflows

You should be tolerant to the decision−making actions that are not perfect but still make progress towards the goal at beginning.
Be strict to the execution action, if the action match the 'KEY FILTERING PRIORITIES' should be filtered.
Especially focus on run_command action, create_file action and edit_file action and their arguments. ( For example, is 'from_pretrained' in the action? What's the observation of them)

Remember: False = Filter out this training turn, True = Keep this training turn

---

**Tool prompt for action-level supervision control**

The judgment result of each turn. The key is the 'turn_id', the value is True or False, representing whether the turn should be kept for training data (True) or filtered out (False).

The 'judge_results' should judge each turn in the history. The 'turn_id' is 'turn {i}', 'i' is a string type number. (The turn is lower case and there is a space between the word turn and the number) The context inside the context between [Start of Turn i] and [End of Turn i] represents the turn's 'i' context.

UNDERSTANDING ACTION TYPES:
Actions fall into two categories that should be evaluated differently:

∗∗DECISION−MAKING ACTIONS∗∗: Planning, reasoning, choosing strategies, deciding what to do next
∗∗EXECUTION ACTIONS∗∗: Actually performing tasks like running commands, training models, file operations

EVALUATION FOCUS:
– ∗∗For Decision−Making∗∗: Evaluate the reasoning quality, planning logic, and strategic thinking
– ∗∗For Execution∗∗: Evaluate actual implementation quality, error handling, and concrete results
– ∗∗Both Types∗∗: Must demonstrate autonomous problem−solving rather than following user directions

The output of each turn should be a single boolean value representing whether to KEEP (True) or FILTER OUT (False) this training example.

---

# G   THE USE OF LARGE LANGUAGE MODELS

We use LLMs to polish our writing, including summarizing long paragraphs to match ICLR's requirements. We also use it to generate the table, but the author double-checks all the data in the table.

