# OpenReview forum: "ARGO: Asynchronous Rollout with Human Guidance for Research Agent Optimization"
_ICLR.cc/2026/Conference — ICLR 2026 Conference Withdrawn Submission_

### Official Review · Reviewer_b832 · 2025-10-27

**Soundness:** 3
**Presentation:** 3
**Contribution:** 2
**Rating:** 4
**Confidence:** 3

**Summary:**

This paper presents ARGO (Asynchronous Rollout with Human Guidance for Research Agent Optimization), a framework aimed at improving long-horizon LLM agent training. ARGO allows human annotators to provide asynchronous, high-level interventions instead of continuous supervision and integrates an action-level supervision control module to mask suboptimal actions, reducing error propagation. Experiments on InnovatorBench demonstrate large improvements over baseline models, suggesting that asynchronous human-in-the-loop sampling can enhance agent learning efficiency and robustness.

**Strengths:**

1. The asynchronous human-guidance mechanism combined with action-level masking represents a creative and practical approach for improving long-horizon agent training.
2. The paper tackles a core problem in the development of autonomous LLM agents — reducing annotation cost while maintaining learning quality — which is highly relevant to the ICLR community.
3. The reported improvements on InnovatorBench and the supporting ablation studies demonstrate meaningful performance gains and validate the importance of human-guided rollouts.

**Weaknesses:**

1. The effectiveness of ARGO appears to depend heavily on the base model’s intrinsic capabilities, the task domain, and the expertise level of annotators. This introduces uncertainty about the method’s generalizability. Moreover, the framework lacks sufficient theoretical guidance or analysis explaining when and why asynchronous human guidance leads to stable learning outcomes across different settings.
2. Several key components — including the masking algorithm, the summarization operator, and the human feedback process — are insufficiently detailed. Without clearer algorithmic definitions and validation experiments, it is difficult to assess reproducibility or understand the exact mechanisms driving the reported gains.

**Questions:**

1. How sensitive is ARGO’s performance to the annotators’ expertise and intervention frequency?
2. Can the authors provide theoretical or empirical evidence on why asynchronous feedback stabilizes optimization?
3. How would ARGO perform on different task types (e.g., GUI automation or multi-modal reasoning)?
4. How robust is the method when base models vary in size or reasoning ability?

---

> ### Author Response · Authors · 2025-11-20
>
> Thanks for the reviewer's comments. Here are our responses to the comments.
>
> **Comment 1:** The effectiveness of ARGO appears to depend heavily on the base model’s intrinsic capabilities, the task domain, and the expertise level of annotators. This introduces uncertainty about the method’s generalizability. Moreover, the framework lacks sufficient theoretical guidance or analysis explaining when and why asynchronous human guidance leads to stable learning outcomes across different settings.
>
> **Response to comment 1:** We believe that we need more and more expert annotators to annotate the tasks when the task is really hard and long horizon, which is what OpenAI, Google DeepMind, and Meta are doing now. If we want to solve such tasks, we just need expert annotators, and these companies can spend money to hire them, so we believe generalizability is not a problem.
>
> **Comment 2:** Several key components — including the masking algorithm, the summarization operator, and the human feedback process — are insufficiently detailed. Without clearer algorithmic definitions and validation experiments, it is difficult to assess reproducibility or understand the exact mechanisms driving the reported gains.
>
> **Response to comment 2:** We have a clear explanation in Section 2 and Appendix D, E. Could you please give us more details about which part you think is insufficiently detailed?
>
> **Comment 3:** How sensitive is ARGO’s performance to the annotators’ expertise and intervention frequency?
>
> **Response to comment 3:** In most cases, human annotators may spend a few seconds to check the log, but in some long-tail cases, they may spend a few minutes (or 10+ minutes) to co-debug with the agent. It's hard for us to estimate the time spent by all human annotators in reading the logs, but the author also annotated some of the tasks, and the time spent is reasonable. (In the beginning 1 hours, it may take for 3-5 minutes on average, but after that, it may take the annotator about 10-30 seconds every hour on average for an agent, because the annotators will not annotate when they are sleeping or away from the computer.)
>
> **Comment 4:** Can the authors provide theoretical or empirical evidence on why asynchronous feedback stabilizes optimization?
>
> **Response to comment 4:** We found that ARGO is more patient and more likely to iteratively refine its outcome, which may stabilize the performance. **This case study is updated in Section 3.5.**
>
> **Comment 5:** How would ARGO perform on different task types (e.g., GUI automation or multi-modal reasoning)?
>
> **Response to comment 5:** Our work mostly focuses on the tasks that need a really long time to finish, for example, it may take 2 days to submit a result. We only found that the AI research domain has such tasks and benchmarks. Also, all of the annotators are from the AI research domain, so it's easy to find related annotators.
>
> **Comment 6:** How robust is the method when base models vary in size or reasoning ability?
>
> **Response to comment 6:** Most of the base models will gain 0 points in this hard task, so it's hard to compare the performance of other base models. And our infrastructure can't support the training of Kimi-K2, so we can't compare the performance of other models.

---

> > ### Comment · Reviewer_b832 · 2025-11-27
> >
> > Thank you for your detailed responses. However, after reviewing them, I still have some conerns with the overall practicality and overhead of the ARGO framework. and I would like to remain my original score. Thanks.

---

### Official Review · Reviewer_iRNb · 2025-11-01

**Soundness:** 2
**Presentation:** 3
**Contribution:** 2
**Rating:** 2
**Confidence:** 4

**Summary:**

The paper introduces ARGO (Asynchronous Rollout with Human Guidance for Research Agent Optimization), a framework designed to train LLM-based research agents for long-horizon, domain-specialized tasks. ARGO enables asynchronous human guidance, where annotators provide high-level interventions only when the agent deviates from a promising trajectory, instead of continuous supervision.

This process is combined with an action-level supervision control mechanism that filters out incorrect or unreliable actions before optimization. The system includes a Human–AI Interaction Interface that allows asynchronous annotation with low cognitive load, integrating visualization, state monitoring, and trajectory control.

ARGO is evaluated on InnovatorBench, using GLM-4.5 as the base model. It achieves over 50% improvement compared to the untrained baseline and 28% improvement over the non-interactive variant. Ablation studies demonstrate that both asynchronous guidance and action-level masking are critical for robust gains, and test-time scaling experiments show that ARGO maintains performance improvements over extended training durations.

**Strengths:**

1.	New Framework Design – ARGO’s combination of asynchronous human oversight and step-level filtering offers a practical, scalable alternative to fully supervised or purely outcome-based training methods. The asynchronous annotation paradigm effectively reduces human cost while maintaining supervision quality.
2. Empirical Validation – Experiments on InnovatorBench show substantial performance gains over both open- and closed-source baselines, including Claude 4 and GPT-5, particularly in domains such as Data Collection, Data Filtering, and Loss Design.
3.	Clear Component Analysis – The ablation study clearly demonstrates performance degradation when removing either the human interaction component or the action-level masking mechanism, validating the necessity of each part.
4.	Scalability and Realism – The Human–AI Interaction Interface is designed for asynchronous updates and minimal cognitive load, making it feasible for long-term, complex research workflows.
5.	Interpretability and Transparency – The explicit action masking and trajectory visualization promote interpretability and enable humans to understand, verify, and improve model behavior efficiently.

**Weaknesses:**

1.	Lack of Theoretical Guarantees – Although conceptually inspired by MDP and ReAct paradigms, ARGO lacks a formal analysis of convergence or stability under asynchronous human guidance.
2.	Limited Evaluation Scope – Experiments are limited to InnovatorBench, without external or cross-domain benchmarks (e.g., GUI or embodied reasoning tasks), which restricts claims of generalization.
3.	Annotation Cost Analysis Missing – While the framework emphasizes reduced human cost, no quantitative comparison of human effort or annotation time is provided.
4.	Potential Evaluation Bias – Human-in-the-loop supervision may introduce domain-specific heuristics that unintentionally align with InnovatorBench’s evaluation metrics, creating potential bias.
5.	Insufficient Discussion of Failure Cases – The paper lacks qualitative examples of failure scenarios, such as incorrect masking decisions or ineffective human interventions.

**Questions:**

1.	How does ARGO handle inconsistent or noisy human feedback during asynchronous rollout? Is there a filtering or weighting mechanism to mitigate contradictory supervision?
2.	What is the measured reduction in annotation time or cost compared with dense, fully supervised baselines?
3.	Can ARGO generalize to multimodal or embodied agents, and how would asynchronous feedback be implemented in such environments?
4.	How is latency or synchronization delay between agent outputs and human responses addressed to prevent drift in long rollouts?
5.	Beyond symbolic masking, are there plans to quantitatively estimate action reliability using uncertainty or self-consistency signals?

---

> ### Author Response · Authors · 2025-11-20
>
> Thanks for the reviewer's comments. Here are our responses to the comments.
>
> **Comment 1:** Lack of Theoretical Guarantees – Although conceptually inspired by MDP and ReAct paradigms, ARGO lacks a formal analysis of convergence or stability under asynchronous human guidance.
>
> **Response to comment 1:** We found that ARGO is more patient and more likely to iteratively refine its outcome, which may stabilize the performance. **A case study is updated in Section 3.5.**
>
> **Comment 2:** Limited Evaluation Scope – Experiments are limited to InnovatorBench, without external or cross-domain benchmarks (e.g., GUI or embodied reasoning tasks), which restricts claims of generalization.
>
> **Response to comment 2:** Our work mostly focuses on the tasks that need a really long time to finish, for example, it may take 2 days to submit a result. We only found that the AI research domain has such tasks and benchmarks. Also, all of the annotators are from the AI research domain, so it's easy to find related annotators.
>
> **Comment 3:** Annotation Cost Analysis Missing – While the framework emphasizes reduced human cost, no quantitative comparison of human effort or annotation time is provided.
>
> **Response to comment 3:** In most cases, human annotators may spend a few seconds to check the log, but in some long-tail cases, they may spend a few minutes (or 10+ minutes) to co-debug with the agent. It's hard for us to estimate the time spent by all human annotators in reading the logs, but the author also annotated some of the tasks, and the time spent is reasonable. (In the beginning 1 hours, it may take for 3-5 minutes on average, but after that, it may take the annotator about 10-30 seconds every hour on average for an agent, because the annotators will not annotate when they are sleeping or away from the computer.)
>
> **Comment 4:** Potential Evaluation Bias – Human-in-the-loop supervision may introduce domain-specific heuristics that unintentionally align with InnovatorBench’s evaluation metrics, creating potential bias.
>
> **Response to comment 4:** If such domain-specific heuristics are general, we believe it's not a problem. What's more, we have assured there is a gap between the training set and the InnovatorBench.
>
> **Comment 5:** Insufficient Discussion of Failure Cases – The paper lacks qualitative examples of failure scenarios, such as incorrect masking decisions or ineffective human interventions.
>
> **Response to comment 5:** **We have added a case study in Section 3.5.**
>
> **Comment 6:** How does ARGO handle inconsistent or noisy human feedback during asynchronous rollout? Is there a filtering or weighting mechanism to mitigate contradictory supervision?
>
> **Response to comment 6:** No. However, we only select expert annotators to annotate the tasks, so the annotators are more likely to give feedback that they think is correct.
>
> **Comment 7:** How is latency or synchronization delay between agent outputs and human responses addressed to prevent drift in long rollouts?
>
> **Response to comment 7:** As we mentioned in the response to comment 3, we sped up the annotation process from 1 hour / agent hours to 10-30 seconds / agent hours. (because the annotators should not leave the computer if they are annotating the task synchronously.)
>
> **Comment 8:** Can ARGO generalize to multimodal or embodied agents, and how would asynchronous feedback be implemented in such environments?
>
> **Response to comment 8:** When the agent have the ability to handle the multimodal or embodied tasks that should cost 2 days, **we believe ARGO can generalize to multimodal or embodied agents**. The annotator can only watch the agent's behavior and talk to them via the chat interface.
>
> **Comment 9:** How is latency or synchronization delay between agent outputs and human responses addressed to prevent drift in long rollouts?
>
> **Response to comment 9:** The annotators are given the whole history when they annotate, but they only need to read the most recent information. Most of the time, they only need to check that the training process is running smoothly. They may need to co-debug with ARGO in some cases, but it may only appear less than 4 times for each task. We found that **some of the annotators would like to chat with the agent to get what problem the agent is facing** (for example, "tell me the problem you are facing and the corresponding files & terminals via thought tool"), and this makes the annotation's cognitive load much lower.
> What's more, we believe this may become a job in the future, so the annotators should remember some of the context to lead this process.
>
> **Comment 10:** Beyond symbolic masking, are there plans to quantitatively estimate action reliability using uncertainty or self-consistency signals?
>
> **Response to comment 10:** No, they are not in our plan.

---

### Official Review · Reviewer_3ibk · 2025-11-06

**Soundness:** 2
**Presentation:** 3
**Contribution:** 3
**Rating:** 4
**Confidence:** 4

**Summary:**

This paper introduces ARGO, a novel training framework for large language model (LLM) agents designed to tackle long-horizon, domain-specialized tasks such as automated research workflows. Traditional approaches like behavior cloning (requiring dense human annotations) and outcome-driven sampling (suffering from sparse rewards) are either too costly or unstable in such settings. ARGO addresses these issues by combining asynchronous human guidance with action-level supervision control, enabling efficient and stable training with minimal human effort. Its contributions are as follows:
1. An asynchronous human guidance mechanism that allows annotators to intervene only when the agent deviates from a promising trajectory, reducing annotation burden.
2. An action-level supervision control strategy that filters out low-quality or erroneous actions, improving training stability and preventing error propagation.
3. A human-AI interaction interface designed for low cognitive load and fine-grained control, making long-horizon annotation practical.
4. Comprehensive experiments on InnovatorBench, showing that ARGO improves over the untrained baseline by >50% and over a non-interactive variant by 28%.

**Strengths:**

S1: The paper identifies a real and important problem in training LLM agents for complex, long-horizon tasks.

S2: The combination of asynchronous guidance and action filtering is intuitive and technically sound.

S3: This paper includes ablation studies, test-time scaling analysis, and comparisons with both open-source and closed-source models.

S4: The human-AI interface is thoughtfully designed to support real-world annotation workflows.

S5: ARGO outperforms baselines across multiple task categories, especially in data collection, filtering, and loss design.

**Weaknesses:**

W1: While effective, the core ideas (human-in-the-loop guidance and action filtering) are not entirely new; the contribution is more of a careful engineering integration.

W2: Tasks are limited to AI research workflows; generalization to other domains (e.g., medicine, law) is not explored.

W3: The impact of annotator expertise or consistency is not studied, which could affect reproducibility and scalability.

W4: Although asynchronous, the system still relies on human judgment at critical points; unclear how it scales to more complex or frequent interventions.

W5: Missing comparisons with stronger baselines: No comparison with RL-based or multi-agent training methods, which limits understanding of ARGO’s relative advantage. Also, apart from using GLM-4.5 as the base model, how about the performance of other base models?

W6: There are some minor typos. For example, Line 112 is an action-level instead of a action-level;  Line 289 is slime code instead of smile code.

**Questions:**

Q1: How does the system handle inconsistent or suboptimal human feedback? Is there a mechanism to down-weight or correct such inputs?

Q2: How is "trajectory deviation" determined? Is it automated, or does it fully rely on human judgment?

Q3: Could the action filtering mechanism be too aggressive, potentially removing exploratory but useful actions?

Q4: Are there plans to test ARGO on non-research tasks or domains outside of AI development?

Q5: Could future versions replace human annotators with learned critics or other agents?

---

> ### Author Response · Authors · 2025-11-20
>
> Thanks for the reviewer's comments. Here are our responses to the comments.
>
> **Comment 1:** While effective, the core ideas (human-in-the-loop guidance and action filtering) are not entirely new; the contribution is more of a careful engineering integration.
>
> **Response to comment 1:** **As far as we know, there are no other papers that have integrated human guidance with the real rollout process asynchronously**. We believe this is a significant contribution to the field of LLM agentic training, since the rollout becomes longer and longer, people can't afford to watch the rollout process all the time. And thanks for the reviewer's support for our engineering efforts.
>
> **Comment 2:** Tasks are limited to AI research workflows; generalization to other domains (e.g., medicine, law) is not explored.
>
> **Response to comment 2:** Our work mostly focuses on the tasks that need a really long time to finish, for example, it may take 2 days to submit a result. We only found that the AI research domain has such tasks and benchmarks. Also, all of the annotators are from the AI research domain, so it's easy to find related annotators.
>
> **Comment 3:** The impact of annotator expertise or consistency is not studied, which could affect reproducibility and scalability.
>
> **Response to comment 3:** In a really hard task, unprofessional annotators are not allowed to annotate the task (or it will only give nonsense feedback). We believe the company will find professional annotators if the task is really important and hard.
>
> **Comment 4:** Although asynchronous, the system still relies on human judgment at critical points; unclear how it scales to more complex or frequent interventions.
>
> **Response to comment 4:**  In most cases, human annotators may spend a few seconds to check the log, but in some long-tail cases, they may spend a few minutes (or 10+ minutes) to co-debug with the agent. It's hard for us to estimate the time spent by all human annotators in reading the logs, but the author also annotated some of the tasks, and the time spent is reasonable. (In the beginning 1 hours, it may take for 3-5 minutes on average, but after that, it may take the annotator about 10-30 seconds every hour on average for an agent, because the annotators will not annotate when they are sleeping or away from the computer.)
>
> **Comment 5:** Missing comparisons with stronger baselines: No comparison with RL-based or multi-agent training methods, which limits understanding of ARGO’s relative advantage.
>
> **Response to comment 5:** We only have 20 training set tasks, so it's hard to do RL. We do have trying multi-agent training method, but we found the judge agent can not give proper feedback to the policy agent, which makes the rollout score in the training set even lower than the baseline (GLM-4.5 without human guidance). (This is different from the action-level masking mechanism, because its context has the whole observation history.)
>
> **Comment 6:** Apart from using GLM-4.5 as the base model, how about the performance of other base models?
>
> **Response to comment 6:** Most of the base models will gain 0 points in this hard task, so it's hard to compare the performance of other base models. And our infrastructure can't support the training of Kimi-K2, so we can't compare the performance of other models.
>
> **Comment 7:** There are some minor typos.
>
> **Response to comment 7:** Thanks for the reviewer's suggestions; we have corrected the typos.
>
> There is an overlap between the questions and weaknesses; we have already answered these questions in the weaknesses section.

---

### Official Review · Reviewer_9Yzq · 2025-11-07

**Soundness:** 3
**Presentation:** 3
**Contribution:** 3
**Rating:** 4
**Confidence:** 3

**Summary:**

This paper proposes ARGO, an asynchronous human-integrated training framework, which includes asynchronous human guidance to construct better trajectory for agent training. After rolling out a trajectory with human guidance, an LLM will generate a summary trajectory based on the raw trajectory and the bad actions will be masked during training.

**Strengths:**

1. Integrating human guidance especially asynchronous guidance is an important yet underexplored topic.
2. The benchmark performance is competitive even compared to closed-source models

**Weaknesses:**

1. The methodology part is a bit unclear. Are the annotators given the whole history when they annotate? This also seems to impose a huge cognitive load. They may need to reload the information each time they annotate, especially if they have been away for a long time.
2. Lack of analysis to demonstrate why asynchronous human guidance is superior compared to other types of human guidance.
3. A fairer comparison is needed. The budget for human annotators should also be considered, and the budget spent by LLMs should match that of humans.
4. typos:
- caption of Figure 2 "originla" -> "original"
-  line 289 smile -> slime
-  line 429 hugh -> huge

**Questions:**

1. How much effort do the annotators put into the experiments? I think a human baseline would be to measure it (e.g., time spent) and compare it to a post-hoc trajectory rewrite using the same amount of effort.
2. Is there any mechanism to reduce the cognitive load for human annotators? For example, can users/annotators chat with the LLM about the current status? It seems to be the case by looking at the screenshots but I'm not so sure.

---

> ### Author Response · Authors · 2025-11-20
>
> Thanks for the reviewer's comments. Here are our responses to the comments.
>
> **Comment 1:** The methodology part is a bit unclear. Are the annotators given the whole history when they annotate? This also seems to impose a huge cognitive load. They may need to reload the information each time they annotate, especially if they have been away for a long time.
>
> **Response to comment 1:** The annotators are given the whole history when they annotate, but they only need to read the most recent information. Most of the time, they only need to check that the training process is running smoothly. They may need to co-debug with ARGO in some cases, but it may only appear less than 4 times for each task. We found **some of the annotators would like to chat with the agent to get what problem the agent is facing** (for example, "tell me the problem you are facing and the corresponding files & terminals via thought tool"), and this makes the annotation's cognitive load much lower.
> Moreover, we believe this may become a job in the future, so the annotators should remember some of the context to lead this process.
>
> **Comment 2: ** Lack of analysis to demonstrate why asynchronous human guidance is superior compared to other types of human guidance.
>
> **Response to comment 2:** If we use synchronous human guidance, such as constantly monitoring the agent’s progress and frequently asking about its status, this would be very tiring for annotators because a single experiment’s sampling process often takes about a full day, during which the agent can move in the right direction without human guidance for long stretches of time. In contrast, asynchronous human guidance only requires annotators to periodically check in on the agent’s progress, determine whether the agent has encountered any problems, and provide solutions if needed, which is much more efficient.
>
> **Comment 3:** A fairer comparison is needed. The budget for human annotators should also be considered, and the budget spent by LLMs should match that of humans.
>
> **Response to comment 3:** In most cases, human annotators may spend a few seconds to check the log, but in some long-tail cases, they may spend a few minutes (or 10+ minutes) to co-debug with the agent. It's hard for us to estimate the time spent by all human annotators in reading the logs, but the author also annotated some of the tasks, and the time spent is reasonable. (In the beginning 1 hours, it may take for 3-5 minutes on average, but after that, it may take the annotator about 10-30 seconds every hour on average for an agent, because the annotators will not annotate when they are sleeping or away from the computer.)
>
> **Comment 4:** Typo corrections.
>
> **Response to comment 4:** Thanks for the reviewer's suggestions; we have corrected the typos.
>
> **Comment 5:** How much effort do the annotators put into the experiments? I think a human baseline would be to measure it (e.g., time spent).
>
> **Response to comment 5:** About 30 seconds to 5 minutes per hour.
>
> **Comment 6:** Compare the time cost to a post-hoc trajectory rewrite using the same amount of effort.
>
> **Response to comment 6:** Actually, it's hard for an annotator to rewrite the trajectory because it can't predict the real response of the environment. Reasoning the response of the environment is even harder than co-debugging with the agent in the process. (And this means the trajectory after that turn is useless.)
>
> **Comment 7:**  Is there any mechanism to reduce the cognitive load for human annotators? For example, can users/annotators chat with the LLM about the current status? It seems to be the case by looking at the screenshots, but I'm not so sure.
>
> **Response to comment 7:** Yes, users/annotators can just ask for the current status of the agent, and the agent will generate the response to the annotator by using the thought tool. Also, the file reader and terminal reader on the right and the search bar on the left can reduce the cognitive load for annotators.

---

> ### Author Response · Authors · 2025-11-20
>
> Thanks for the reviewer's comments. Here are our responses to the comments.
>
> **Comment 1:** The methodology part is a bit unclear. Are the annotators given the whole history when they annotate? This also seems to impose a huge cognitive load. They may need to reload the information each time they annotate, especially if they have been away for a long time.
>
> **Response to comment 1:** The annotators are given the whole history when they annotate, but they only need to read the most recent information. Most of the time, they only need to check that the training process is running smoothly. They may need to co-debug with ARGO in some cases, but it may only appear less than 4 times for each task. We found **some of the annotators would like to chat with the agent to get what problem the agent is facing** (for example, "tell me the problem you are facing and the corresponding files & terminals via thought tool"), and this makes the annotation's cognitive load much lower.
> Moreover, we believe this may become a job in the future, so the annotators should remember some of the context to lead this process.
>
> **Comment 2: ** Lack of analysis to demonstrate why asynchronous human guidance is superior compared to other types of human guidance.
>
> **Response to comment 2:** If we use synchronous human guidance, such as constantly monitoring the agent’s progress and frequently asking about its status, this would be very tiring for annotators because a single experiment’s sampling process often takes about a full day, during which the agent can move in the right direction without human guidance for long stretches of time. In contrast, asynchronous human guidance only requires annotators to periodically check in on the agent’s progress, determine whether the agent has encountered any problems, and provide solutions if needed, which is much more efficient.
>
> **Comment 3:** A fairer comparison is needed. The budget for human annotators should also be considered, and the budget spent by LLMs should match that of humans.
>
> **Response to comment 3:** In most cases, human annotators may spend a few seconds to check the log, but in some long-tail cases, they may spend a few minutes (or 10+ minutes) to co-debug with the agent. It's hard for us to estimate the time spent by all human annotators in reading the logs, but the author also annotated some of the tasks, and the time spent is reasonable. (In the beginning 1 hours, it may take for 3-5 minutes on average, but after that, it may take the annotator about 10-30 seconds every hour on average for an agent, because the annotators will not annotate when they are sleeping or away from the computer.)
>
> **Comment 4:** Typo corrections.
>
> **Response to comment 4:** Thanks for the reviewer's suggestions; we have corrected the typos.
>
> **Comment 5:** How much effort do the annotators put into the experiments? I think a human baseline would be to measure it (e.g., time spent).
>
> **Response to comment 5:** About 30 seconds to 5 minutes per hour.
>
> **Comment 6:** Compare the time cost to a post-hoc trajectory rewrite using the same amount of effort.
>
> **Response to comment 6:** Actually, it's hard for an annotator to rewrite the trajectory because it can't predict the real response of the environment. Reasoning the response of the environment is even harder than co-debugging with the agent in the process. (And this means the trajectory after that turn is useless.)
>
> **Comment 7:**  Is there any mechanism to reduce the cognitive load for human annotators? For example, can users/annotators chat with the LLM about the current status? It seems to be the case by looking at the screenshots, but I'm not so sure.
>
> **Response to comment 7:** Yes, users/annotators can just ask for the current status of the agent, and the agent will generate the response to the annotator by using the thought tool. Also, the file reader and terminal reader on the right and the search bar on the left can reduce the cognitive load for annotators.

---

### Note · Authors · 2026-01-04

I have read and agree with the venue's withdrawal policy on behalf of myself and my co-authors.